# The Effectiveness of Spirituality-Centered Cognitive Therapy on Body Image, Sexual Function, Illness Perception and Intrusive Thoughts in Iranian Women after Mastectomy

**Mehdi Sharifi** [1], **Harold G. Koenig** [2], **Mahboubeh Dadfar** [3,*], **Yahya Turan** [4] **and Alireza Ghorbani** [5]

[1] Psychology Department, Islamic Azad University, Bandar Gaz 48731-97179, Iran; mehdisharifi273@gmail.com
[2] Departments of Psychiatry and Medicine, Duke University Health System, Durham, NC 27705, USA; harold.koenig@duke.edu
[3] Addiction Department, Iran University of Medical Sciences, Tehran 14496-14535, Iran
[4] Islamic Sciences, Bandırma Onyedi Eylül University, Bandırma 10200, Türkiye; yahyaturans@gmail.com
[5] Social Sciences, Payam Noor University, Tehran 14556-42183, Iran; it_ghorbani@yahoo.com
[*] Correspondence: mahboubehdadfar@yahoo.com or dadfar.m@iums.ac.ir

**Abstract:** Spirituality-centered cognitive therapy refers to the way in which people search for and express the meaning and purpose of their lives, as well as experience connection with themselves, others, nature, and spirituality. This study aimed to determine the effectiveness of spirituality-centered cognitive therapy on body image, sexual function, disease perception, and disturbing thoughts in women after mastectomy. This was semi-experimental research, which was conducted via a pre-test–post-test method and had a control group. The samples included 85 women with breast cancer who had undergone mastectomy, and, based on the inclusion criteria, 78 individuals were randomly selected and were then divided into two groups (39 individuals in the intervention group and 39 individuals in the control group). The intervention group received eight 120 min sessions of spirituality-centered cognitive therapy, and the control group did not receive any training. Data were collected using questionnaires on illness perception, body image, sexual function, and rumination and were then analyzed by multivariate analysis of variance with repeated measurements using SPSS-24. Before the training, there was no significant difference between the intervention and control groups in the scores obtained by the scales. After the intervention, the mean scores in all scales except sexual function were significantly different from the control group. Therefore, spirituality-centered cognitive therapy may be useful for improving negative psychological symptoms among women in Iran with breast cancer after mastectomy surgery.

**Keywords:** spirituality-centered cognitive therapy; body image; sexual function; illness perception; intrusive thoughts; breast cancer; mastectomy

## 1. Introduction

Breast cancer refers to a malignant tumor that grows in breast cells. In recent years, this type of cancer has been considered the main cause of cancer-related death in women, with an upward trend in the incidence rate in Iran (Akbari et al. 2017). Although the prevalence of breast cancer in Iran is lower than that in developed countries, an important point that should be noted is the younger age of women with breast cancer in Iran, meaning that a significant number of women of a certain age are affected by this disease and its psychological and social consequences, which has made breast cancer one of the most important challenges in the field of health in Iran (Fazeli et al. 2013). In Iran, the late diagnosis of this disease in more advanced stages due to the higher cost of treatment in more advanced stages of the disease, on the one hand, and due to disruption in patients' jobs, on the other hand, has a high financial burden, which puts a great deal of pressure on the family and the health system (Hakami Shalamzai et al. 2022).

There are various methods such as surgery, chemotherapy, radiotherapy, and hormone therapy to treat breast cancer, which are different based on the degree and severity of the disease. In Iran, due to the lack of screening, breast cancer is usually diagnosed in the advanced stages of the disease, and, therefore, it is logical that 81% of the surgeries performed to treat breast cancer are mastectomies (Tahergorabi et al. 2014). Moreover, many problems in patients with breast cancer are due to the trauma caused by breast removal surgery and its effects on their quality of life. This is because after mastectomy, the patient experiences conditions similar to cutting off other parts of the body—in this case, a part that is a symbol of sex, being a woman, and being a mother—and by removing it, women suffer a great deal of stress and are likely to develop mood disorders such as depression and anxiety, in addition to a reduction in the quality of life (Karataş et al. 2017). Memaryan et al. (2017) stated that after mastectomy, patients suffer from mood disorders such as depression and anxiety. In other studies, it was found that after surgery, the quality of life decreased drastically, and there is a significant relationship between mood disorders and quality of life.

Under cultural and social conditions, breast loss or breast deformity can lead to negative changes in body image and self-concept (Renshaw 1994). Body image is defined as individuals' mental image of their body and the attitude they have towards their body, appearance, health status, natural function, and sexual attractiveness, and it is regarded as one of the most important criteria for character development and results in high self-esteem and a positive self-concept in women (Alinejad Mofrad et al. 2021). According to the results obtained by Pecor (2004), breast removal through surgery is considered the destruction of a part of the body, and in this process, young women are less satisfied with their body image after surgery than older patients. In terms of marital compatibility, the most important aspect for women is their feeling and belief about their femininity, with women often feeling worried about changing their physical and sexual status and their spouses not accepting this problem; therefore, it seems that concern about body image after mastectomy surgeries also affects sexual function (Anagnostopoulos et al. 2010). Previous studies in various countries found that women feel that their sexual function is affected by mastectomy (Molavi et al. 2015; Pirnia et al. 2020). According to Sheydaei Aghdam et al. (2019), whenever the physical and sexual attraction of couples for each other is higher, they are more satisfied with their sexual relations.

Disease perception refers to people's understanding and management of their illness by developing cognitive representations based on their knowledge and previous experiences (Carnelli et al. 2017). Disease perception includes emotional evaluation. A negative perception of the disease leads to anxiety and distress (Cook et al. 2015). It has been reported that the perception of illness plays a role in both experiencing emotional states and using coping strategies (Krok et al. 2019). Disease perception is a major factor regarding the perception and management of cancer and other chronic diseases, which has a significant impact on the patient's emotional response to the disease, compatibility with treatment, and functional health status (Karataş et al. 2017). Considering the type of perception of the illness and its consequences, cancer can cause problems in a person's perception and adaptation to it (Saritas and Özdemir 2018). In the process of adapting to the illness, these patients react to their disease and medical care in a special way (Vatvani et al. 2017). Patients who perceive their illness as curable show better personal control and treatment than those who consider it incurable, and their despair is also lower; this issue reminds us of the need to reduce the perceived threat of the disease and strengthen the beliefs of disease control in patients with cancer (Park et al. 2020). Perceptions of the disease thus influence the mechanisms preferred to cope with the illness.

Theoretically, perceptions of disease serve as a framework for the individual's chosen coping strategies (Aydın Sayılan and Demir Doğan 2020). Also, patients who perceive their illness as serious, chronic, and uncontrollable show worse physical and mental health. It seems that disease perceptions play an important role in perceived health by patients with breast cancer (Nehir et al. 2019). In addition, disease perceptions have a significant

correlation with psychological distress and the emotional well-being of patients with breast cancer (Fischer et al. 2013); therefore, disease perceptions explain a significant part of the variance of psychological well-being in breast cancer (Dempster and McCorry 2012). Based on the research, patients with breast cancer are very interested in making changes in their lives and especially in changing their health habits. The tendency to make these changes seems to be more related to people's differences in disease perceptions. Thus, beliefs about the consequences of the disease, causal factors, and prevention of relapse are related to behavioral changes after treatment (Costanzo et al. 2011).

Cancer can be considered an important stressor in the formation of post-traumatic stress disorder. The sense of self-worth, autonomy, and security decreases when individuals face this life-threatening condition (Torabi et al. 2018). These patients report negative images, anger, inconsistent judgments, avoidance, preoccupation with belief in fate, despair, and feeling helpless and helplessness, which can play an important role in maintaining depression and anxiety in these patients. Moreover, patients report intrusive thoughts, negative images, inconsistent judgments, and feelings of helplessness (Karekla and Constantinou 2010). The most prominent stress-related symptoms reported by patients with cancer are intrusive thoughts about their illness along with efforts to avoid these thoughts, which are prominent features of a stress response syndrome (Renna et al. 2021). Beatty and Koczwara (2010) showed that up to 60% of women with breast cancer suffer from symptoms of post-traumatic stress disorder, such as intrusive thoughts.

In order to improve mental health and reduce psychological problems, many counselors have proposed and developed various psychotherapies such as behavioral therapy, cognitive therapy, and metacognitive therapy, as well as existential, analytical, and integrated therapies (Haaga et al. 1991). The main challenge of the therapies above is that the treatments can sometimes lose their effectiveness, and the symptoms of the disorders return after a short time, with psychological problems and critical states intensifying (Haddadi Kuhsar et al. 2017).

Cognitive behavioral therapy, one of these therapeutic approaches, is a psychotherapy that integrates cognitive and behavioral principles as well as behavioral, cognitive, and rational emotional therapy techniques. The central claim of cognitive behavioral therapy is that thought patterns and beliefs, emotional states, and behaviors are interconnected. Therefore, cognitive behavioral therapy involves two essential efforts to change cognitive processes and behavior accordingly. How individuals feel and behave generally determines their perceptions and comments (Pearce et al. 2015). In other words, cognitive and behavioral therapy aims to transform the wrong beliefs and thoughts of the individual, to increase their incomplete knowledge, to change the way they perceive and interpret events, and to gain a positive attitude as a result (Horne and Watson 2011). In the literature, there is evidence that cognitive and behavioral therapy is an effective method for coping with stressful life events (Groarke et al. 2013; Horne and Watson 2011). Dealing with stressful life events is a dynamic process strongly influenced by individual and cultural factors and can take different forms within the framework of cultural and religious traditions (Coughlin 2008; Henderson et al. 2003). Therefore, adding religious and spiritual elements to cognitive and behavioral therapy will provide significant advantages in impacting the more profound points of the individual's spiritual life and providing a powerful and lasting transformation in perspectives. These issues were among those that brought the attention of thinkers to the spiritual dimension of man and the inclusion of spiritual components in psychological theories, and the positive effect of this orientation was confirmed in many psychological studies (Nasution and Afiyanti 2021).

Many studies have studied the relationship between religion, spirituality, and mental health. Although conflicting findings are reported among various studies, meta-analytical studies reveal that religion and spirituality are influential factors in mental health. They note that spiritual/religious coping can increase pain management, improve surgical outcomes, and protect against depression (Koenig and Larson 2001; Larson and Larson

2003; Yapıcı 2007, pp. 50–51). Studies indicate that religion and religiosity increase positive emotions such as well-being, hope, and optimism (Cohen and Koenig 2004).

It seems that in surgeries in which an organ is removed from the body, control over the body is reduced, and this causes the person to feel incompetent concerning their own body, leading to insecurity and a negative mental image. The breasts are an important part of a woman's self-image, and they are strongly associated with gender identity, sexuality, physical attractiveness, self-confidence, nurturing, and a sense of motherhood. Therefore, after surgery, a number of women worry about a perceived reduction in their feminine attractiveness. Breast removal changes their physical attractiveness schema in their view, which causes sexual and marital satisfaction problems. Due to these maladaptive and distorted thoughts, they lose the ability to regulate their emotions and to adopt favorable coping styles (Tahergorabi et al. 2014). Spiritual therapy helps to identify these maladaptive thoughts and distorted thinking styles and to evaluate them, as well as use spiritual teachings for replacing the thoughts with more adaptive and realistic thoughts (Memaryan et al. 2017). According to this therapeutic approach, connecting with a source beyond the patient's existence, dealing with spiritual themes, and trying to find the meaning of life can lead to providing conditions for recovery after the traumatic event (Pearce et al. 2015). It is also stated in the Quran that peace of mind is possible only by turning to God. In Rad's Surah, "Verily in the remembrance of Allah do hearts find rest" (13:28).

Cancer diagnosis and its treatment are challenging for a woman and may threaten the meaning of their life and sometimes lead to a feeling of collapse (Alinejad Mofrad et al. 2021). Research has shown that spirituality acts as an "inner resource or inner aspect of a person" against a wide range of stressful events that people face (Dadfar et al. 2023; Sharifi et al. 2023). Spiritual coping can help people deal with their problems. The mechanism behind this type of coping may reflect finding meaning, purpose, and hope, which in turn will strengthen people and aid them in dealing with their pain and suffering (Jafari et al. 2013). In this process, spiritual coping is defined as the use of cognitive and behavioral techniques when faced with stressful life events, which arise from a person's spirituality. Other research has shown that patients tend to increase their focus on religion and connection with God as their cancer progresses (Ghahari et al. 2017). Religiosity is also thought to have a protective mechanism in mental health conditions, including against depression (Koenig and Larson 2001; Koenig 2008; Koenig and Büssing 2010). Also, Sajadi et al. (2018) showed that spiritual approaches are the main coping strategy in Iranian patients with cancer, who consider spirituality the primary source of coping and hope. According to Hajabadi et al. (2020), when people—especially Muslims—are affected by chronic or incurable diseases such as cancer, they often report that their religious beliefs and practices are a source of comfort in reducing physical and mental discomfort.

Although much research has been carried out regarding the importance of paying attention to psychological issues in patients with breast cancer, paying attention to psychological issues as a part of treatment and complementing medical treatment has not found its place as it should have. For this reason, these patients can be found to have a low quality of life and with problems in family-related and social fields. On the other hand, despite the existence of studies on the effectiveness of different therapy methods to improve psycho-social and family-related conditions, it is still necessary to use new treatment methods due to existing shortcomings, especially in relation to paying attention to psychological issues.

Therefore, in order to improve the coping methods of patients with cancer, especially in women who had undergone mastectomy, it is necessary to have a more comprehensive understanding of women's experiences and how to modify their coping methods. Therefore, this study aimed to investigate the effect of spirituality-centered cognitive therapy on body image, sexual function, disease perception, and intrusive thoughts in women after mastectomy. There is a rich body of literature on religion and spirituality (e.g., Hill et al. 2000; Spilka and McIntosh 1996; Zinnbauer et al. 1997). This research was conducted in Iran, where Islam is dominant. Religiousness and spirituality have different meanings

and are intertwined in Islam, whether Sunni or Shia. According to Islam, belief in Allah is the first requisite to be a Muslim, and spirituality independent of the core teachings of Islam is not possible. In addition, the development of the spirituality of the individual, according to Islamic thought, will be revealed by the level of compliance with the beliefs and behaviors emphasized in the Quran. Abdulkadiroğlu (1996) asserts that religion itself is a spiritual way of life and is of significant value in the life of human beings, constituting the foremost spiritual element. Additionally, Dr. Robert Emmons suggests that spirituality should be considered a distinct type of human intelligence. Spiritual intelligence fundamentally involves an individual's ability to navigate and discover meaning and significance in the surrounding world. In Islamic theology, this process entails reflecting on the signs (ayat) of Allah present in the world and conveying knowledge to individuals about how to act, think, and feel in accordance with these signs. For example, when a person observes the changes in trees during the autumn season, they perceive it as a sign of Allah (Abdul-Rahman 2017). Islam derives spirituality from religion and considers it an integral and complementary part of religiosity. Therefore, it is highly incorrect to view Sufism, which we will briefly describe as the spiritual or inner life of Islam, separately from Islam. Spirituality is the "essence", while religion is the "form" (Albayrak 2015). In Christianity, religiosity is formally structured and defined by religious beliefs, practices, and community, whereas spirituality is more centered on the individual (who may be religious or not). While in Christianity, religiosity and spirituality are distinguishable, they are not completely independent of one another. Previous research has shown that many participants describe themselves as spiritual but not religious. Among these individuals, there is often a tendency to not be engaged in institutional forms of religion but rather to participate in group experiences related to spiritual development, to have New Age beliefs, and to have mystical experiences (Dadfar et al. 2019).

## 2. Methodology

This was semi-experimental research, which was done by a pre-test–post-test method, and it utilized a control group with a random assignment of subjects. Figure 1 shows the flowchart for this study.

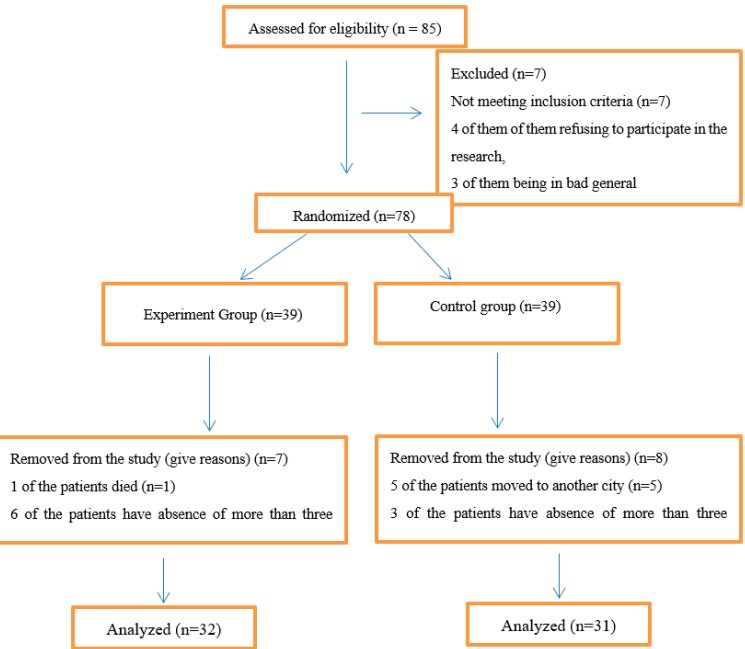

**Figure 1.** Flowchart for this study.

*2.1. Participants*

Sampling occurred between January and May 2021 in the oncology centers of Mashhad in Iran. The samples included 85 women with breast cancer who had undergone mastectomy and, based on the inclusion criteria, 78 individuals were randomly selected and were then divided into two groups (39 individuals in the intervention group and 39 individuals in the control group).

The inclusion criteria for subjects was being between the ages of 30 and 60; the ability to attend meetings and respond to research tools; having a mastectomy (partial or complete); not attending spiritual therapy sessions in the past; being married and in a continuous cohabitation (with wife not undertaking long absences); having experienced at least two years of married life; not having a chronic illness such as psychotic mental disorders, physical and mental disabilities, AIDS, or hepatitis; the absence of addiction to alcohol and drugs; and non-attendance in other therapy sessions during cognitive therapy sessions. The exclusion criteria included being absent from more than three therapy sessions, not being able to perform daily tasks, not suffering from other chronic or serious medical illness, and not using psychiatric drugs.

All the participants were provided with information about this study, and their written informed consent to participate in the research was obtained. Also, in order to comply with the ethical aspects of the research and with the consent of the control group, spirituality-oriented cognitive intervention was held for this group as well at the end of the research.

*2.2. Measures*

2.2.1. The Revised Illness Perception Questionnaire (IPQ-R)

The IPQ-R was developed by Moss-Morris et al. (2002). It is a reliable and valid tool that can be used to diagnose cognitive and emotional representations of illness among different patients, including people with cancer (Dempster and McCorry 2012). This questionnaire has eight subscales, seven of which are cognitive, including the identity of the illness (14 items), the course of the acute-chronic illness (6 items), the course of the periodic illness (6 items), the consequences of the illness (6 items), the therapeutic control of the illness (5 items), personal control over the illness (6 items), coherence (clear understanding of the illness, 5 items), and emotional representations of the illness (6 items) to measure the emotional reactions of patients towards the illness. In addition, it includes four subscales of causal representation, i.e., psychological attributions, risk factors, safety, and randomness or chance. In the identity scale, a yes answer results in a score of one point and a no answer results in a score of zero points. The rest of the scales are scored based on a five-point Likert scale, from completely disagree with a score of 1 to completely agree with a score of 5. High scores in illness identity, an acute-chronic illness course, a periodic illness course, illness consequences, and emotional representations of the illness indicate an unfavorable condition in these scales, and high scores in therapeutic control, personal control, and coherence (a clear understanding of the illness) represents a favorable situation and positive perceptions. In Iran, Basharpoor et al. (2018) reported the Cronbach's alpha coefficient of the Persian version of this questionnaire as above 0.70 on patients with cancer (including breast cancer).

2.2.2. Rumination Scale

This questionnaire was developed by Nolen-Hoeksema and Morrow (1991). The questionnaire has 22 items; the respondents are asked to rate each one on a scale from never (1) to most of the time (4), and the total score of the questionnaire is calculated from the sum of the scores of all the items. Scores can vary between 22 and 88. The higher the score, the higher the rumination level. One study reported the reliability of this scale using Cronbach's alpha between 0.88 and 0.92, and an intraclass correlation of 0.75. The Cronbach's alpha obtained in the Iranian sample is reported to be 0.86 (Tanhaye Reshvanloo et al. 2021).

### 2.2.3. Body Image and Relationship Scale

This scale was developed by Hormes et al. (2008) to measure the body image in women with breast cancer after mastectomy surgery. This scale contains 32 items that measure three subscales, including ability and health (12 items), social complications (9 items), and physical appearance and sexual attractiveness (11 items).

### 2.2.4. Female Sexual Function Index (FSFI-6)

This index is considered to be a standard tool for evaluating the sexual function of women with and without sexual dysfunction, and it has been translated and validated in more than thirty countries. This index evaluates sexual function during the last five weeks in eight areas, including sexual desire, sexual arousal, vaginal moisture, orgasm, pain, and sexual satisfaction, and it aims to provide a concise and accurate scale for screening women for FSD that can be implemented and scored in clinical and research situations. From the original version that has 19 items, a short form of 6 items was developed and validated to measure female sexual function. Each question covers a main issue. For this scale, good internal consistency and reliability (test–retest) have been reported, and it can significantly distinguish women with FSD from women with normal sexual function and satisfaction, which indicates its appropriate diagnostic validity. The psychometric features of the FSFI-6 were investigated in Iranian women, showing that this scale has a good Cronbach's alpha coefficient (0.83), and each of the items showed a good correlation with the total score, signifying the good internal consistency of the FSFI-6 (Molavi et al. 2015).

### 2.2.5. Procedure

The nature of program, treatment goals, and general characteristics of treatment intervention were clarified for the treatment staff working in oncology centers. Then, the staff were asked to provide the researcher with a list of names of patients who had undergone mastectomy surgery. According to the inclusion criteria, 85 eligible patients were invited to participate in the briefing sessions. Then, researchers introduced themselves and explained the research objectives to the patients. Among them, 78 who were willing to participate in the study were divided into two experimental and control groups, and a pre-test was performed for them. However, a total of 15 patients left the study (in the experimental group, 1 patient died, and 6 patients were absent for more than three sessions; and in the control group, 5 patients moved their place of residence to another city, and 3 patients were absent more for than three sessions). Therefore, this study was conducted with 63 patients (32 patients in the intervention group and 31 patients in the control group).

Spirituality-centered cognitive therapy based on multi-dimensional therapy was developed by Dehkhoda (2013). This therapy uses the concepts and techniques of cognitive therapy to correct cognitive errors, maladaptive thoughts, and distorted thinking styles. These errors, thoughts, and styles prevent realistically dealing with existential anxieties, such as the unpredictability and uncontrollability of affairs, death, meaninglessness, and loneliness. Humanistic beliefs and perceptions of God are corrected by helping the person accept suffering and find positive meaning, which in turn will help the patient better adapt. The therapy was performed in 8 one-hour sessions 2 times a week from October to December 2021. Focusing on specific goals for patients, the therapy was conducted in three steps (initial: sessions 1–2; middle: sessions 3–6; and final: sessions 7–8) (see Table 1). Spirituality-centered cognitive therapy has three cognitive, spiritual, and existential components. According to this, maladaptive thoughts and distorted thinking styles were examined in these sessions.

The first stage included providing a better sense of security in a safe space; developing a therapeutic relationship; getting to know each other; informing the patients about the spiritual care program to determine the session rules and help the patients express their thoughts, feelings, and opinions; and explaining cognitive distortions and then challenging them. At this stage, the therapist explained emotional problems, life conditions, and spiritual beliefs. The patients were taught that the key to improve and change their thoughts was the replacement

of negative thoughts with positive ones. As an assignment, the patients were asked to review their negative beliefs about spirituality, God, and cancer for 5–10 min a day according to the number of negative beliefs. Based on the content in this stage, the patients were encouraged to accept and replace negative beliefs with positive beliefs.

**Table 1.** Summary of spirituality-centered cognitive therapy sessions for women after mastectomy.

| Session | Goals |
|---|---|
| 1st Session | - Familiarity with the general goals of the treatment and expressing the group's expectations and goals, as well as place and time of sessions<br>- Revealing primary anxieties |
| 2nd Session | - Requesting the clients to tell their stories about how they contracted the illness and their emotional and behavioral reactions to it<br>- Helping the self-expression and self-discovery of the therapist and to be present here and now |
| 3rd Session | - Facilitating the expression of spiritual beliefs and practices of the therapist<br>- Knowing the therapist's spiritual coping style through evaluating thoughts and rules and challenging them<br>- Correcting the patient's cognitive distortions regarding attributing the cause of her illness to her past sins<br>- Assessing the cause of cancer from the point of view of the patient, understanding the existing cognitive errors, and finally challenging them |
| 4th Session | - Facing the concept of existential anxieties while telling the story of cancer<br>- Dealing with the concept of the unpredictability of the world in the form of the client's fear of relapse or death and the use of her spiritual beliefs in accepting uncertainty<br>- Evaluating the clients' thoughts and assumptions regarding the uncontrollability of cancer-related issues and anxiety caused by it<br>- Challenging mentioned thoughts through cognitive techniques and correcting the distortions hidden in them<br>- Helping clients fulfill their spiritual needs in this regard, such as self-forgiveness and creating honest relationships with themselves, others, and God<br>- Working with the theme of death anxiety when expressing the reaction to a cancer diagnosis<br>- Correcting existing cognitive distortions in the meaning of the existential anxiety of death from the point of view of the therapist<br>- Encouraging clients to participate in spiritual ceremonies<br>- Hope and trust: expressing the concept of hope, the benefits of hope, and hope therapy according to studies |
| 5th Session | - Talking about how to do the homework of the previous session and encouraging the therapist to express her personal meaning derived from the presented spiritual themes and work on them<br>- Helping to create new meaning in the event of possible death by using her spiritual beliefs in order to help accept death as an inevitable reality<br>- Examining the degree of difference between overall meaning in an event and situational meaning (assigned) by the therapist to interpret the experience of the event<br>- Teaching spiritual skills according to the needs of the therapist |
| 6th Session | - Helping to accept the responsibility of giving meaning to life experiences<br>- Helping to re-evaluate priorities and life experiences<br>- Using spiritual beliefs of clients in finding meaning<br>- Helping the therapist to find an efficient meaning free of cognitive distortions |
| 7th Session | - Challenging the concept of suffering<br>- Helping the therapist accept suffering following the process through cognitive techniques<br>- Helping the client delve into her established reality in order to free herself from suffering and accept it |
| 8th Session | - Talking with the client about the identity that is confused due to the dysfunctional meaning process, inconsistent thoughts, etc.<br>- Teaching meditation, prayer, and how to find spiritual security in order to reach peace in times of need<br>- Summing up the previous sessions and helping the client reach the conclusion that behind every experience there is a personal meaning, and a person has unconditional freedom when choosing this meaning<br>- Necessary coordination of follow-up meetings and post-examination implementation |

The middle stage focused on the unpredictability and uncontrollability of death, responsibility, and meaninglessness; correcting the humanistic view of God; and finally accepting suffering via the spiritual needs field derived from other needs. Each session began with a summary of the previous sessions, a review of the assignments, and discussion about the progress of the patients in relation to their new thoughts and feelings that they had achieved after completing the assignments, particularly regarding those concerning a higher being, God, spirituality, and illness. Then, by gathering information about the purposefulness of life, their beliefs were checked with the following questions: Does this disease have the power to impact all aspects of your life? What is your purpose in life? Negative and unhelpful thoughts were identified, and beliefs that lead to negative feelings were challenged.

The final stage was dedicated to talking about the patient's identity that was confused due to a dysfunctional meaning process; inconsistent thoughts; teaching meditation; prayer; how to find spiritual security to reach peace in times of difficulty; widening the patients' horizons about what they could do in life; emphasizing that gratitude can be expressed through different means such as the heart, tongue, and body; summarizing the sessions; and evaluating the effectiveness of the therapy. As part of the therapeutic process, the patients were encouraged to repeat a spiritual word from the Holy Quran in their mind and to pay attention to and focus on the words. This practice helped preserve the thoughts and attitudes that the patients needed to challenge their negative thoughts. The therapist assisted patients in improving their ability to monitor thoughts, to modify negative thoughts, and to replace them with positive ones. The patients also learned how to interpret events and to understand how they affected their mood. Moreover, the patients were taught how to choose alternative ways to respond to negative beliefs and expectations based on their value systems and personal goals. In this regard, it was emphasized that engaging in spiritual acts such as praying and meditating could help them develop favorable coping responses to negative events.

## 3. Analysis

Data analysis was carried out using descriptive statistics (mean and standard deviation), Box's M test, the Levine test, and multivariate analysis of variance with repeated measurements using SPSS-24. The significance level used for the results was $p < 0.05$.

## 4. Results

Table 2 shows the demographic characteristics of the subjects. As indicated by the results presented in Table 3, the mean of the body image variables, illness perception, and intrusive thoughts in the experimental group in the post-test stage shows a difference compared to the pre-test stage (Table 3).

**Table 2.** Demographic characteristics of both groups.

| Variables | | Control (n = 31) | | Experiment (n = 32) | |
|---|---|---|---|---|---|
| | | **n** | **%** | **n** | **%** |
| Marital status | Married | 21 | 67.8 | 24 | 75.0 |
| | single | 10 | 32.2 | 8 | 25.0 |
| Education | Elementary school | 8 | 25.8 | 5 | 15.6 |
| | High School | 11 | 35.5 | 12 | 37.5 |
| | BA | 8 | 25.8 | 10 | 31.3 |
| | MA and above | 4 | 12.9 | 5 | 15.6 |

**Table 2.** *Cont.*

| Variables | | Control (n = 31) | | Experiment (n = 32) | |
|---|---|---|---|---|---|
| | | n | % | n | % |
| Work status | Full-time | 9 | 29.0 | 7 | 21.9 |
| | Part-time | 5 | 16.2 | 6 | 18.8 |
| | Unemployed | 17 | 54.8 | 19 | 59.3 |
| Level of income | Good | 11 | 35.5 | 8 | 25.0 |
| | Medium | 15 | 48.3 | 20 | 62.5 |
| | Bad | 5 | 16.2 | 4 | 12.5 |
| Having children | Yes | 16 | 51.6 | 18 | 56.3 |
| | No | 15 | 48.4 | 14 | 43.7 |
| Treatment time | 6 months–2 years | 7 | 22.6 | 9 | 28.1 |
| | 3–5 years | 24 | 77.4 | 23 | 71.9 |

**Table 3.** Mean and standard deviation of pre-test and post-test scores of variables.

| Groups | Experiment | | | | Control | | | | Follow-Up | |
|---|---|---|---|---|---|---|---|---|---|---|
| | Pre-Test | | Post-Test | | Pre-Test | | Post-Test | | | |
| Variables | M | SD | M | SD | M | SD | M | SD | M | SD |
| Body Image | 52.3 | 7.28 | 56.78 | 5.10 | 50.64 | 5.40 | 51.10 | 4.83 | 56.78 | 5.10 |
| Sexual Function | 35.40 | 11.3 | 38.01 | 9.21 | 36.11 | 8.78 | 37.01 | 9.10 | 34.58 | 12.90 |
| Illness Perception | 44.28 | 3.73 | 56.7 | 2.10 | 45.3 | 1.32 | 46.9 | 1.08 | 55.7 | 2.17 |
| Intrusive Thoughts | 61.25 | 9.18 | 48.09 | 6.34 | 59.90 | 5.80 | 60.39 | 4.58 | 44.61 | 8.27 |

Table 4 shows the results of the multivariate analysis of variance with repeated measures for intergroup and intragroup effects of dependent variables. The scores of at least one of the dependent variables of the research in the three times of pre-test, post-test and follow-up have a significant difference ($p < 0.001$). Also, the interaction between groups (experiment and control) and time (pre-test, post-test, and follow-up) was observed in at least one of the dependent variables (with a significant difference of $p < 0.001$).

**Table 4.** The results of multivariate analysis of variance with repeated measures for intergroup and intragroup effects of dependent variables.

| Effect | | Test | Value | Hypoth. DF | Error. df | F | Sig |
|---|---|---|---|---|---|---|---|
| Between-Group | Group | Wilks' lambda | 0.66 | 8 | 21 | 3.67 | 0.001 |
| | Time | Wilks' lambda | 0.29 | 16 | 93 | 9.36 | 0.001 |
| Intragroup | Interaction of time and group | Wilks' lambda | 0.13 | 16 | 93 | 8.78 | 0.001 |

Table 5 shows the results of an analysis of variance with repeated measurements of the scores of dependent variables in three stages: pre-test, post-test, and follow-up. In all dependent variables except sexual function at the pre-test and post-test levels, there is a significant difference in the post-test and follow-up levels ($p < 0.001$ and $p < 0.05$). Therefore, the intervention maintained its sustainability over time in the experimental group.

**Table 5.** The results of analysis of variance with repeated measurement of the scores of the dependent variables in the pre-test, post-test, and follow-up stages.

| Source | Variable | Stage | Sum of the Squares | df | The Mean of the Squares | F | Sig | The Effect Size |
|---|---|---|---|---|---|---|---|---|
| Time | Body image | Pre-test–Post-test | 341.11 | 1 | 341.11 | 22.24 | 0.00 | 0.48 |
| | | Post-test–follow-up | 0.63 | 1 | 0.63 | 2.71 | 0.07 | 0.17 |
| | Sexual Function | Pre-test–Post-test | 501.2 | 1 | 501.2 | 38.65 | 1.00 | 0.11 |
| | | Post-test–follow-up | 0.002 | 1 | 0.002 | 0.00 | 0.52 | 0.13 |
| | Illness Perception | Pre-test–Post-test | 353.11 | 1 | 353.11 | 30.21 | 0.00 | 0.44 |
| | | Post-test–follow-up | 0.58 | 1 | 0.58 | 2.72 | 0.00 | 0.64 |
| | Intrusive Thoughts | Pre-test–Post-test | 67.23 | 1 | 67.23 | 3.19 | 0.04 | 0.33 |
| | | Post-test–follow-up | 13.22 | 1 | 13.22 | 0.01 | 0.00 | 0.19 |

Table 6 shows the results of variance analysis with repeated measurements of scores of dependent variables in the three stages of pre-test, post-test, and follow-up in the experimental and control groups. In all the dependent variables except sexual function at the pre-test and post-test levels, there is a significant difference in the post-test and follow-up levels in the experimental and control groups ($p < 0.001$ and $p < 0.05$). Therefore, the intervention maintained its sustainability over time in the experimental group. Also, Bonferroni's post hoc test of pairwise comparisons showed that in the post-test stage, compared to the pre-test stage, the mean scores of the dependent variables of body image, illness perception, and intrusive thoughts in the experimental group compared to the control group changed significantly. In the follow-up stage, compared to the post-test stage, the average scores of the dependent variables in the experimental group did not change significantly compared to the control group.

**Table 6.** The results of analysis of variance with repeated measurements of the scores of dependent variables in the pre-test, post-test, and follow-up stages in the experimental and control groups.

| Source | Variable | Stage | Sum of the Squares | df | The Mean of the Squares | F | Sig | The Effect Size |
|---|---|---|---|---|---|---|---|---|
| Interaction of time and group | Body image | Pre-test–Post-test | 528.00 | 1 | 528.00 | 30.11 | 0.00 | 0.51 |
| | | Post-test–follow-up | 15.13 | 1 | 15.13 | 10.43 | 0.08 | 0.19 |
| | Sexual Function | Pre-test–Post-test | 203.24 | 1 | 203.24 | 25.31 | 0.78 | 0.19 |
| | | Post-test–follow-up | 0.78 | 1 | 0.78 | 12.09 | 1.00 | 0.23 |
| | Illness Perception | Pre-test–Post-test | 312.56 | 1 | 312.56 | 31.09 | 0.00 | 0.56 |
| | | Post-test–follow-up | 0.18 | 1 | 0.18 | 6.70 | 0.00 | 0.23 |
| | Intrusive Thoughts | Pre-test–Post-test | 61.76 | 1 | 61.76 | 4.13 | 0.00 | 0.19 |
| | | Post-test–follow-up | 17.65 | 1 | 17.65 | 1.09 | 0.01 | 0.13 |

## 5. Discussion

The results showed that spirituality-centered cognitive therapy had a positive effect on illness perception in women after mastectomy, which is in line with Alinejad Mofrad et al. (2021), Haddadi Kuhsar et al. (2017), and Sheydaei Aghdam et al. (2019). Memaryan et al. (2017) reported that spiritual health can be effective in improving the perception of illness in all subscales in women with breast cancer. Spirituality-centered cognitive therapy, through increasing the sense of positivity towards events, increasing hope and trust in God, and accepting everything that God has destined for a person, reduced distress in patients with breast cancer and, therefore, enabled them to tolerate the illness and

pain, easing the complications caused by the mastectomy. What is important in spiritual therapy is its semantic aspect, which forms the content of therapy and is manifested in the form of worship in every religion. These techniques in spiritual therapy can have behavioral, cognitive, metacognitive, emotional, and moral aspects and are manifest in religious concepts and behaviors such as trust, patience, prayer, meditation, etc.

Spirituality-centered cognitive therapy has also been effective on body image and intrusive thoughts in women with breast cancer after mastectomy surgery. Since the breasts are an important part of a woman's body image, any type of abnormality in the breasts can lead to a negative body image in a woman. Patients who undergo a mastectomy experience body image problems because the surgery makes the patient feel like they have lost an important part of their femininity. Disease perception refers to people's understanding and management of their illness by developing cognitive representations based on their knowledge and previous experiences (Carnelli et al. 2017). Disease perception includes an emotional evaluation. A negative perception of the disease leads to anxiety and distress (Cook et al. 2015). It was reported that the perception of illness plays a significant role in experiencing emotional states and using coping strategies (Krok et al. 2019). In this regard, in spirituality-centered cognitive therapy, patients are taught to re-examine their incompatible thoughts and replace them with more compatible and realistic views through communication with a source beyond their existence and addressing spiritual themes and efforts to find the meaning of life. Allah states in the Quran that He tests people with goodness and evil (Quran, Surat al-Araf, 168; Surat al-Baqara, 155). Believers are asked to be patient in the face of calamities (Quran, Hajj Surah 35). Allah says in verse 156 of Surat al-Baqara, "Those who, when afflicted by a calamity, say: Our entire existence belongs to Allah, and Him, we return." In other words, establishing a solid bond with the Almighty Creator and showing submission are among the spiritual values of Islam. The individual who enters into the mind-set of deficiency and dysfunction after mastectomy surgery thinks that she has lost social acceptance and appreciation and has low self-esteem (Mete and Beydağ 2021), but she can reorganize her incompatible thoughts within the framework of the perspective expressed in the Quran during the spirituality-centered therapy process.

According to the results, the role of spiritual therapy in improving sexual function was not significant. Because, in addition to correcting beliefs and cognitive errors in women after mastectomy and since this operation targets one of the most important female sexual organs, this issue can affect marital relationships. Therefore, if the husband does not play a supporting role towards the patient and does not help her in accepting this issue, spiritual therapy cannot help in this case. Also, in these women, constant mental preoccupation with the perceived defect may cause low sexual excitability or even a lack of excitability. Frederickson and Robert point out that when people are distracted by their appearance, they cannot focus on their sexual pleasure, and this factor has a negative effect on their sexual function. In line with this, Holzner et al. (2001) showed that patients with breast cancer often had sexual dysfunction. Hannoun-Levi (2005) and Onen Sertoz et al. (2004) found that mastectomy and chemotherapy lead to negative changes in mental image, decreases in self-esteem and sense of femininity, and sexual disorders, especially in young women. Schoenberg (1979) believes that patients after mastectomy consider themselves as a failed and rejected person and avoid having sex with their spouse. Such behavior causes a feeling of rejection in the husband, such that he stops contacting her. At this stage, the woman also experiences a feeling of being pushed back, and a defective cycle is created based on the lack of a proper and satisfying relationship. One of the factors that may cause women to have sex with their husbands after mastectomy is sexual consideration. In sexual consideration, a person sees the needs of the other party as a higher priority than his own needs. The consideration performed by the patient's husband can be impaired for reasons such as the patient's physical weakness and the perception of sexual intercourse as a health-threatening factor, while the consideration performed by the patient is due to two main reasons, including worrying about her husband's extramarital affairs and respecting his needs. According to the experimental study conducted by Brandberg et al. (2008),

sexual pleasure before surgery and one year after had a statistically significant difference, which was consistent with the results of this study.

The results of this research and our ability to generalize the findings to other populations are affected by several limitations. First, the research participants were all Muslims, which limits our ability to generalize the results to followers of other religions. Second, this intervention was performed in 5 weeks on women with breast cancer after mastectomy. Nevertheless, this study has a number of strengths, including that the principles of clinical trials, including random allocation, were followed in this study. The prospective nature of this research allows for the observation of cause and effect, which is often not possible in purely correlational studies. One of limitations of this study was the absence of counseling for the control group.

Another limitation of this study is that a test measuring the spirituality levels of the participants was not included in the research. This study focused on the factors of body image, sexual function, disease perception, and disturbing thoughts that could negatively affect the mental health of women who had mastectomy surgery. The effect of spirituality-centered cognitive therapy was analyzed by assuming that the current spirituality levels of the research group were equal. It is recommended that in similar studies to be conducted in the future, spirituality levels should also be determined, and the effects of spirituality-centered cognitive therapy on individuals with different spirituality levels should also be analyzed.

In this study, spirituality-centered cognitive therapy was applied to the experimental group, while no counseling was given to the control group in order to prevent any placebo effect. Other studies to be conducted after this study could shed light on finding the most effective method by comparing different therapy applications.

## 6. Conclusions

This paper provides evidence that spirituality-centered cognitive therapy can be used as a complementary and alternative treatment modality, as a health intervention for the psychological management of patients with breast cancer after mastectomy surgery. The findings of this study indicated that intrusive thoughts in the experimental group decreased significantly in the post-test stage compared to the pre-test stage, highlighting the potential effectiveness of spiritual therapy in reducing distressing thoughts. Apart from sexual activity, spirituality-centered cognitive therapy can also have a healing effect on body image and the perception of illness, which was supported by the results of this study. Moreover, this study found that spiritual therapy maintained its effect in the post-intervention follow-up period, indicating that spirituality and religion have the power to transform distorted cognitions and prevent intrusive thoughts, especially for religious individuals. It is recommended that healthcare professionals include spiritual care programs as part of patient education programs and consider this approach as a valuable tool in the psychosocial care of patients with breast cancer. Additionally, psychological counseling and support should be provided to patients after mastectomy surgery, and group discussions should be facilitated so that patients can talk with others who have experienced similar situations and express their concerns and attitudes. Spiritual therapy training can be a complementary treatment option for patients with breast cancer, and it does not lose its effect over time, as it takes its power from the individual's world of belief. In conclusion, incorporating spiritual and psychosocial care into breast cancer treatment can have significant benefits for patients, and healthcare professionals should consider the use of spiritual therapy as an important part of the overall care plan for patients with breast cancer after mastectomy surgery.

**Author Contributions:** Conceptualization, M.S., H.G.K., Y.T. and A.G.; methodology, M.S. and A.G.; software, M.S.; validation, M.S., M.D. and H.G.K.; formal analysis, M.S.; investigation, M.D.; resources, A.G.; data curation, A.G.; writing—original draft preparation, M.S., H.G.K., M.D. and A.G.; writing—review and editing, M.D. and Y.T.; visualization, Y.T.; supervision, H.G.K.; project

administration, M.S. and A.G.; funding acquisition, M.S. All authors have read and agreed to the published version of the manuscript.

**Funding:** This research received no external funding.

**Institutional Review Board Statement:** Not applicable.

**Informed Consent Statement:** Not applicable.

**Data Availability Statement:** Derived data supporting the findings of this study are available from the corresponding author on reasonable request.

**Conflicts of Interest:** The authors declare no conflict of interest.

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
