# Peer review of "The Effectiveness of Spirituality-Centered Cognitive Therapy on Body Image, Sexual Function, Illness Perception and Intrusive Thoughts in Iranian Women after Mastectomy"

_religions, doi:10.3390/rel15050578_

Round 1

Reviewer 1 Report (Previous Reviewer 2)

Comments and Suggestions for Authors

The author/s have significantly improved their paper according to my suggestions.

Author Response

Reviewer 1

Comments and Suggestions for Authors

The author/s have significantly improved their paper according to my suggestions.

Author’s Respond: Thank you for your feedback.

Reviewer 2 Report (New Reviewer)

Comments and Suggestions for Authors

This is a quite good and interesting study (doing psychological research outside the WEIRD sample is always a pro) but I find some serious methodological flaws that would need to be revised:

1. "Due to these religious and 212
cultural differences, religion and spirituality are used together in the article, without mak- 213
ing a distinction between the two." (p. 5) => this is a bad idea! both in religious studies and in psychology religiosity and spirituality are defined in many ways. There are authors like Hood Skrzypińska or Socha that would oppose the authors' assumption and say that the term spirituality IS DIFFERENT from religiosity...

2. Both Christianity and Islam are treated as coherent entities WHICH IS NOT TRUE. that is a common methodological mistake that is made in psychological studies. At least it should be noted which Islamic tradition (Iran is a hint here) people were involved in.

3. the title is VERY MISLEADING - it is a sine qua non condition that the sample in question should be involved in the title. this is no general study on a mass population but it is done in specific cultural environment. it is also a sine qua non condition for me to revise that paper again;

I do not read the article further cause these flaws need to be corrected or adressed in the first place.

Comments on the Quality of English Language

It is ok. minor revision.

Author Response

Reviewer 2

This is a quite good and interesting study (doing psychological research outside the WEIRD sample is always a pro) but I find some serious methodological flaws that would need to be revised:

  1. "Due to these religious and cultural differences, religion and spirituality are used together in the article, without making a distinction between the two." (p. 5) => this is a bad idea! both in religious studies and in psychology religiosity and spirituality are defined in many ways. There are authors like Hood Skrzypińska or Socha that would oppose the authors' assumption and say that the term spirituality IS DIFFERENT from religiosity...

Author’s Respond: Between lines 134-149, information about spirituality in the literature is given. Additionally, the explanation made earlier is expanded between lines 198-226.

  1. Both Christianity and Islam are treated as coherent entities WHICH IS NOT TRUE. that is a common methodological mistake that is made in psychological studies. At least it should be noted which Islamic tradition (Iran is a hint here) people were involved in.

Author’s Respond Shia Islamic understanding is expressed in line 202.

  1. the title is VERY MISLEADING - it is a sine qua non condition that the sample in question should be involved in the title. this is no general study on a mass population but it is done in specific cultural environment. it is also a sine qua non condition for me to revise that paper again;

Author’s Respond: A sample has been added to the title.

This manuscript is a resubmission of an earlier submission. The following is a list of the peer review reports and author responses from that submission.

Round 1

Reviewer 1 Report

Comments and Suggestions for Authors

Please check that the authors listed in the manuscript are included in the references.  And also delete any bibliographic references not appearing in the manuscript. Please check the names for the transliteration of Arabic to Latin characters. They should always appear the same.

What appears in pag 5, is Beshrpour et alii (2017) and in the reference section may be it Basharpoor … (2018).

What appears in pag 5, is Nolen-Hoeksema and Maro (1991) and in the reference section there is not their entry.

What appears in pag 5, is Jurman (2006) and in the reference section there is not this entry.

What appears in pag 5, is Rashvanlou et alii (2021) and in the reference section there is not their entry.

In what concerns the Rumination Scale Jurman and Rashvanlou entries are needed to verify the validity criterion in cross cultural psychology, that is, the scale translated measures what is intended in the original version.

What appears in pag 5, is Horms et alli (2008) and in the reference section there is not this entry and, if available, it allows to verify the validity criterion, that is, the scale translated measures what is intended in the original version.

Please, insert at least another paragraph in the comment for each table 3, 4, 5 and 6. We write articles so that the reader understands what each table entails with reliable data. The purpose is to highlight the data in support of what we audit. As written and presented, it appears to be a passing annotation in every quoted table.

Author Response

Review Report (Reviewer 1)

Comments and Suggestions for Authors

Please check that the authors listed in the manuscript are included in the references.  And also delete any bibliographic references not appearing in the manuscript. Please check the names for the transliteration of Arabic to Latin characters. They should always appear the same.

 Responds to Comments and Suggestions of Reviewer 1

What appears in pag 5, is Beshrpour et alii (2017) and in the reference section may be it Basharpoor … (2018). (done)

What appears in pag 5, is Nolen-Hoeksema and Maro (1991) and in the reference section there is not their entry. (done)

What appears in pag 5, is Horms et alli (2008) and in the reference section there is not this entry and, if available, it allows to verify the validity criterion, that is, the scale translated measures what is intended in the original version. (just added reference)

Reviewer 2 Report

Comments and Suggestions for Authors

I have carefully reviewed the manuscript, titled “The Effectiveness of Spirituality-centered Cognitive Therapy 2 on Body Image, Sexual Function, Illness Perception and Intrusive Thoughts in Women after Mastectomy”. The study was to examine the effectiveness of spirituality-centered cognitive therapy on body image, sexual function, disease perception and disturbing thoughts in women after mastectomy.

The study has been substantially revised and some strong points (clear introduction, accurate statistical analysis, constructive discussion).

However, I would like to ask the authors to address some points in order to improve the paper

Introduction:

1) Can you present some psychological explanations for spirituality? Why is it so importance in terms of therapy in general? What are its underlying mechanisms (p. 1-2)

2) As the study examines associations between illness perception and the cognitive factors in cancer patients, it would be beneficial to provide more information on the role of illness perception, which is included in the following works:

https://onlinelibrary.wiley.com/doi/full/10.1002/pon.5157?casa_token=Vg4f_p2mhCgAAAAA%3AoPq0VFYMgyameV_8Z7aswuBS4BQ1C6c9A6mJHU2jKkIVbOCVt5H_VuQXTDugU5KYvtXqnB0alLU7pQ (Illness perception and affective symptoms in gastrointestinal cancer patients: A moderated mediation analysis of meaning in life and coping)

https://onlinelibrary.wiley.com/doi/full/10.1111/ecc.13252?casa_token=eIzftUwzxzgAAAAA%3ANU7_fUMTR9UO8sMrRqE0IK3UZXvt-rf3bzJg9_GvfktE97AE1V6pJAbMP9i5VWA5URV851OYdyI1Lw (Illness perception, perceived social support and quality of life in patients with diagnosis of cancer)

https://www.tandfonline.com/doi/abs/10.1080/10508619.2018.1556061 (When Meaning Matters: Coping Mediates the Relationship of Religiosity and Illness Appraisal with Well-Being in Older Cancer Patients)

Method:

3) Was the sample determined by power analysis?

4) How did you handle missing values in your data? (If any exist)

Results:

5) The results are properly presented.

Discussion:

6) What are the underlying mechanisms responsible for this result: “Spirituality-centered cognitive therapy has also been effective on body image and intrusive thoughts in women with breast cancer after mastectomy surgery.” (p. 10)?

7) Can you elaborate on the following statement: “Patients who had undergone mastectomy experience body image problems because the surgery makes the patient feel like they have lost an important part of their femininity (p. 10)”. Please, provide a potential explanation of this result.

8) In Limitations there should be more information regarding the specific role  of the spirituality measurement used in the study.

Author Response

Review Report Reviewer 2

Comments and Suggestions for Authors

I have carefully reviewed the manuscript, titled “The Effectiveness of Spirituality-centered Cognitive Therapy 2 on Body Image, Sexual Function, Illness Perception and Intrusive Thoughts in Women after Mastectomy”. The study was to examine the effectiveness of spirituality-centered cognitive therapy on body image, sexual function, disease perception and disturbing thoughts in women after mastectomy.

 The study has been substantially revised and some strong points (clear introduction, accurate statistical analysis, constructive discussion).

 However, I would like to ask the authors to address some points in order to improve the paper

Responds to Comments and Suggestions of Reviewer 2

Introduction:

Can you present some psychological explanations for spirituality? Why is it so importance in terms of therapy in general? What are its underlying mechanisms (p. 1-2) (done)

As the study examines associations between illness perception and the cognitive factors in cancer patients, it would be beneficial to provide more information on the role of illness perception, which is included in the following works:

https://onlinelibrary.wiley.com/doi/full/10.1002/pon.5157?casa_token=Vg4f_p2mhCgAAAAA%3AoPq0VFYMgyameV_8Z7aswuBS4BQ1C6c9A6mJHU2jKkIVbOCVt5H_VuQXTDugU5KYvtXqnB0alLU7pQ (Illness perception and affective symptoms in gastrointestinal cancer patients: A moderated mediation analysis of meaning in life and coping) (done)

https://onlinelibrary.wiley.com/doi/full/10.1111/ecc.13252?casa_token=eIzftUwzxzgAAAAA%3ANU7_fUMTR9UO8sMrRqE0IK3UZXvt-rf3bzJg9_GvfktE97AE1V6pJAbMP9i5VWA5URV851OYdyI1Lw (Illness perception, perceived social support and quality of life in patients with diagnosis of cancer) (done)

https://www.tandfonline.com/doi/abs/10.1080/10508619.2018.1556061 (When Meaning Matters: Coping Mediates the Relationship of Religiosity and Illness Appraisal with Well-Being in Older Cancer Patients) (done)

 Discussion:

What are the underlying mechanisms responsible for this result: “Spirituality-centered cognitive therapy has also been effective on body image and intrusive thoughts in women with breast cancer after mastectomy surgery.” (p. 10)? (done)

Can you elaborate on the following statement: “Patients who had undergone mastectomy experience body image problems because the surgery makes the patient feel like they have lost an important part of their femininity (p. 10)”. Please, provide a potential explanation of this result. (done)

Reviewer 3 Report

Comments and Suggestions for Authors

The article presents a fascinating study based on Spiritual Cognitive Behavioral Therapy. The authors attempt to explore the effectiveness of such a treatment approach through quasi-experimental research. However, the article falls short in two critical areas.

Firstly, it lacks a clear theoretical and methodological foundation to support the study. (Pearce MJ, Koenig HG, Robins CJ, Nelson B, Shaw SF, Cohen HJ, King MB brought a great contribution to this novel approach. See: “Religiously integrated cognitive behavioral therapy: a new method of treatment for major depression in patients with chronic medical illness”. Psychotherapy (Chic). 2015 Mar;52(1):56-66. doi: 10.1037/a0036448).

Secondly, a clear protocol for assessing the effectiveness of the therapy sessions is also absent.

The article provides a detailed account of the ten therapy sessions; however, there needs to be a clear indication of how maladaptive thoughts and distorted thinking styles were collected and integrated into the sessions.

In addition, several statements demonstrate confusion about the role of the client and the therapist: “Correcting the existing cognitive distortions in the meaning of the existential anxiety of death from the point of view of the therapist”; “Teaching spiritual skills according to the needs of the therapist”; “Helping the therapist to find an efficient meaning free of cognitive distortions”; “Helping the therapist to accept the suffering following the meaning process, through cognitive techniques”.

In sum, its lack of a theoretical and methodological foundation and the absence of a clear protocol for evaluating the effectiveness of the therapy undermines the study's findings.

Author Response

Review Report (Reviewer 3)

Comments and Suggestions for Authors

 The article presents a fascinating study based on Spiritual Cognitive Behavioral Therapy. The authors attempt to explore the effectiveness of such a treatment approach through quasi-experimental research. However, the article falls short in two critical areas.

Responds to Comments and Suggestions of Reviewer 3

Firstly, it lacks a clear theoretical and methodological foundation to support the study. (Pearce MJ, Koenig HG, Robins CJ, Nelson B, Shaw SF, Cohen HJ, King MB brought a great contribution to this novel approach. See: “Religiously integrated cognitive behavioral therapy: a new method of treatment for major depression in patients with chronic medical illness”. Psychotherapy (Chic). 2015 Mar;52(1):56-66. doi: 10.1037/a0036448). (done)

Round 2

Reviewer 3 Report

Comments and Suggestions for Authors

Nothing was altered in the text regarding my comments in the round 1.
My observations remain the same.

Author Response

Thank you very much for your review. Please check the attachment.
